# Predictors of Growth of Vestibular Schwannoma After Gamma Knife Treatment: A Systematic Review

**DOI:** 10.3390/cancers17121993

**Published:** 2025-06-14

**Authors:** Cheng Yang, Daniel Alvarado, Pawan Kishore Ravindran, Max E. Keizer, Koos Hovinga, Martinus P. G. Broen, Danielle Eekers, Inge Compter, Henricus P. M. Kunst, Yasin Temel

**Affiliations:** 1Department of Neurosurgery, Maastricht University Medical Center, 6202 AZ Maastricht, The Netherlands; 2Dutch Academic Alliance Skull Base Pathology, Maastricht University Medical Center, 6202 AZ Maastricht, The Netherlands; 3Department of Neurology, Maastricht University Medical Center, 6202 AZ Maastricht, The Netherlands; 4Department of Radiation Oncology (MAASTRO), Maastricht University Medical Center, 6202 AZ Maastricht, The Netherlands; 5Department of Otorhinolaryngology, Maastricht University Medical Center, 6202 AZ Maastricht, The Netherlands; 6Department of Otorhinolaryngology, Radboud University Medical Center, 6525 GA Nijmegen, The Netherlands; 7Dutch Academic Alliance Skull Base Pathology, Radboud University Medical Center, 6525 GA Nijmegen, The Netherlands; 8Istanbul Atlas University, 34406 Istanbul, Turkey

**Keywords:** systematic review, vestibular schwannoma, growth rate, Gamma Knife, skull base, factors

## Abstract

Gamma Knife radiosurgery (GKRS) effectively controls vestibular schwannoma growth, yet some tumors still progress after treatment. This systematic review analyzed predictors of post-GKRS growth, including patient demographics, tumor characteristics, and imaging biomarkers. We found that initial tumor volume, pretreatment growth rate, and MRI features were significant predictors of regrowth. Understanding these factors may improve personalized treatment strategies and outcomes for patients undergoing GKRS for vestibular schwannoma.

## 1. Introduction

Representing the predominant benign neoplasm within the adult cerebellopontine angle (CPA), vestibular schwannomas (VS, acoustic neuromas) constitute more than 80% of lesions in this region [1]. These tumors develop from the Schwann cells responsible for myelinating the vestibular component of the eighth cranial nerve. Although these tumors are histologically benign, their growth within the confined space of the CPA can cause compression of adjacent structures, including the cochlear and facial nerves, as well as the brain stem. Clinically, ipsilateral sensorineural hearing impairment is the most prevalent symptom, affecting over 90% of patients. Additionally, dizziness or disturbances in balance occur in approximately 61% of cases, while asymmetric tinnitus occurs in roughly 55% of individuals [2]. Recent studies suggest that the lifetime prevalence of sporadic VS may exceed 1 in 500 people [3].

With the improvement of magnetic resonance imaging technology, VS are now being diagnosed at much earlier stages [4]. The treatment method is usually determined based on the size, morphology, and symptoms of the tumor. For growing asymptomatic small- to medium-sized tumors, stereotactic radiosurgery (SRS) has become an increasingly utilized primary treatment due to the high tumor control rate [5]. Literature reports control rates exceeding 94% [6]. Gamma Knife radiosurgery (GKRS) is a form of SRS. It targets tumors with precisely focused high-dose radiation, aiming to inhibit their growth while maximizing the protection of surrounding normal tissue. Despite its effectivity, a minority of tumors grow after GKRS treatment, presenting a significant challenge for patient management. However, only a few studies have explored the factors influencing tumor growth after GKRS, and their findings are often inconsistent. This limits clinicians’ ability to develop personalized treatment plans for patients.

Here, we provide a review of the literature on the growth rate of sporadic VS after GKRS, evaluating existing evidence on clinical, anatomical, and imaging-based predictors, with the aim of identifying the factors that influence this growth.

## 2. Materials and Methods

The protocol for this review can be found in the PROSPERO online database of systematic reviews (ID: CRD42024604277).

### 2.1. Search Strategy

This systematic review was performed in accordance with the Preferred Reporting Items for Systematic Reviews and Meta-Analyses (PRISMA) guidelines. PubMed, EMBASE, and Cochrane were searched using the following search terms: ((Acoustic Neuroma) OR (vestibular schwannoma)) AND (growth) AND (radiotherapy). The search period is from 1 January 2000 to 1 January 2024. Following the export of search results into EndNote (Clarivate Analytics), duplicate records were removed. Two authors (C.Y. and D.A.) independently screened the remaining articles based on title and abstract. Publications deemed relevant at this stage underwent full-text assessment by the same authors. Articles meeting the predefined inclusion criteria during this detailed appraisal were selected for inclusion. To ensure methodological consistency, a third author (Y.T.) independently validated this selection process. Any discrepancies arising between reviewers were resolved through consensus discussion involving all three researchers.

### 2.2. Study Selection

The inclusion criteria were determined according to the PICOS method (population, issue of interest, comparison, outcome and study design) [7]. The population (P) was the patients with VS after GKRS; the issue of interest (I) was tumor progression or regrowth following GKRS; the comparison (C) was the stable or regressing tumors; the outcome (O) was the growth rate of the VS after GKRS; and the study designs (S) were retrospective cohort studies. Additional inclusion criteria applied if they: (1) were peer-reviewed original research articles reporting outcomes in VS patients treated with GKRS, (2) contained data on post-GKRS tumor growth rate, and (3) were published in English. Studies were excluded if they were: (1) Studies involving patients with neurofibromatosis type 2 (NF2), (2) cohorts including individuals with prior VS surgical or radiological treatment, (3) reviews, meta-analyses, case reports, comments, books, information pages, or animal or phantom studies, or (4) publications in languages other than English.

### 2.3. Data Extraction

Two authors (C.Y. and D.A.F.) independently extracted study data from articles. The third author (Y.T.) subsequently validated all extracted information to ensure accuracy. The following parameters were systematically recorded: authors and publication year, number of participants, gender, age, tumor control outcomes, radiation dose, mean follow-up time, complications, and prognostic factors. Missing data were defined as data that were either not directly reported or if the required data could not be indirectly extrapolated, if data or specific prognosis factors were not reported in an included study, we marked the corresponding cells in the data extraction table with a forward slash (‘/’) to indicate not mentioned. To prevent double-reporting patients in different publications by the same research group, the publication with more detailed and abundant data was used.

### 2.4. Quality Evaluation

Based on the significant clinical and methodological heterogeneity found during the initial data extraction and study assessment phase, pooling such heterogeneous data may produce misleading results; thus, we did not conduct a meta-analysis and used a qualitative analysis approach. We qualitatively assessed heterogeneity, sensitivity, and potential bias among studies.

Heterogeneity was assessed descriptively, focusing on treatment protocols and growth outcome definitions. Risk of bias was assessed using the Newcastle-Ottawa Scale (NOS), which assesses study quality in terms of selection, comparability, and outcome dimensions [8]. Studies were evaluated using eight questions designed to assess their quality, with a maximum possible score of 9. Only those publications that achieved a total score of 7 or higher were classified as high quality, indicating satisfactory performance across evaluation criteria. Sensitivity was qualitatively addressed by excluding studies with NOS scores < 5 and avoiding the inclusion of overlapping patient cohorts reported by the same study group. To enhance the robustness of the study results, only moderate- to high-quality (NOS ≥ 6) studies were included. Two authors (C.Y. and D.A.F.) independently evaluated the quality of each study. A third reviewer (Y.T.) was designated to make a final decision if the initial two reviewers were unable to reach consensus.

### 2.5. Statistical Analysis

Statistical analysis was conducted using SPSS V.28 (IBM SPSS Statistics, IBM Corp., Armonk, New York, NY, USA) to perform descriptive statistics, including means, ranges, and proportions for patient characteristics, tumor control rates, and prognostic factors.

## 3. Results

### 3.1. Characteristics of Inclusion in the Study

Our search initially generated three hundred and sixty-one articles for review, including three hundred and thirty-one from PUBMED, twenty-seven from EMBASE, and three from Cochrane. As depicted in Figure 1, a total of nine out of the three hundred and sixty-one articles were deemed suitable for this study (see Figure 1). Across the nine included studies, all were retrospective in design and single-center in scope, with publication years ranging from 2011 to 2023. All studies utilized Gamma Knife radiosurgery exclusively, though some differed in specific planning systems and immobilization protocols. Notably, tumor control definitions and imaging modalities varied significantly. Some studies relied on volumetric measurement using segmentation software, while others used simple linear measurements.

This figure shows the flowchart of the literature search, with a total of 361 articles retrieved from PubMed, Cochrane, and Embase databases. After removing duplicate articles and screening according to the inclusion and exclusion criteria, 270 articles were obtained. Screening the titles and abstracts of each result excluded 242 irrelevant articles. A full-text review of the remaining 28 articles in detail excluded an additional 19 articles, as they were unrelated to tumor growth rate. The remaining nine articles were included in the study.

### 3.2. Quality Assessment

Significant clinical and methodological heterogeneity was observed across studies. This primarily stemmed from variations in the definition of tumor progression and radiation protocols. All nine included articles were evaluated: eight were considered high-quality studies, and one was considered moderate quality (Appendix A). No studies with a NOS score below 5 were included. Although we were unable to perform a statistical assessment of publication bias, it is notable that five out of the nine included studies reported negative findings for the primary predictors. This indicates a lower risk of selective reporting and suggests that publication bias may be limited in this review.

### 3.3. Overall Findings

A total of 1964 cases with unilateral VS treated with GKRS were included (see Table 1). There was no evident predilection for sex (973 female and 991 male participants), and the ages of the included subjects ranged between 18 and 95 years. Subjects were given an average marginal dose of 12.5 Gy (range: 8.3–20.0 Gy). The follow-up period ranged between 6 and 240 months. Successful radiological control was achieved in 92.9% of the subjects. In the remaining one hundred and thirty-nine subjects where no control was achieved, forty-seven chose to repeat radiation therapy, thirty-six chose microsurgical resection, and five chose a wait-and-scan strategy. The salvage treatment of the remaining 51 patients was not reported.

### 3.4. Patient Factors

The prognostic significance of patient-specific factors such as age, gender, and administered radiation dose remains uncertain in predicting post-treatment outcomes for VS. Out of the included studies, eight investigated the correlation between post-GKRS tumor growth and age; however, no significant correlation was reported [9,10,12,13,14,15,16,17]. Gender has likewise shown no significant influence on post-treatment tumor growth. Variations in marginal radiation dose administered, although crucial for immediate therapeutic effects, have not consistently predicted long-term tumor growth control.

### 3.5. Tumor Factors

Two studies reported a positive correlation between higher post-GKRS control rate and smaller initial tumor size. Stijn et al. [17] showed significantly higher 5-year actuarial control rates for tumors < 0.5 cm^3^ (94.1%) compared to those ≥0.5 cm^3^ (90.0%), with further stratification showing superior control in tumors < 6 cm^3^ (92.2%) versus ≥6 cm^3^ (80.7%). Similarly, Stephen et al. [13] also identified improved control rates for tumors < 0.56 cm^3^. However, in the other five studies [10,11,14,15,16], no significant relationship between the initial size and post-treatment regrowth was reported.

Divergent findings were observed by three studies regarding the prognostic significance of pretreatment tumor growth kinetics. Marston et al. [11] reported significantly higher control rates for tumors with baseline growth rates < 2.5 mm/year (*p* = 0.007). Conversely, Soroush Larjani et al. [16] noted a non-significant trend where rapidly growing tumors exhibited greater growth rate reduction post-GKRS. Meanwhile, Wangerid et al. [12] found no predictive value of pretreatment growth rates.

Anatomical factors, such as tumor location classified by the Koos grading system, have also been examined. Among six studies [9,10,11,12,13,14] assessing tumor location, five [9,10,12,13,14] concluded that tumor growth was not significantly related to its anatomical position. An exception was Marston et al. [11], who identified extra-canalicular (EC) extension as a predictor of post-GKRS progression. Regarding morphology, Chih-Chun Wu et al. [15] demonstrated that cystic VSs exhibited a larger volume reduction ratio after GKS compared to solid tumors, translating to a lower regrowth rate. In contrast, three studies [9,14,17] reported there is no correlation between solid and cystic tumors.

### 3.6. Clinical Presentation

Despite the importance of clinical features in the initial diagnosis, their prognostic value in predicting tumor regrowth after GKRS remains unclear. One study reported this variable. According to Fernidand C. A. Timmer et al. [10], there is no significant correlation between the initial clinical symptoms and the likelihood of tumor recurrence or progression following GKRS treatment.

### 3.7. Apparent Diffusion Coefficient (ADC)

A study by Chih-Chun Wu [15] (n = 187) investigated ADC values as a predictive biomarker. Pre-GKRS ADCmax values were significantly higher in patients with stable/regressing tumors compared to those with progression (*p* = 0.010). Similarly, a significantly higher 6-month post-GKRS ADCmean measurements were seen in stable/regressing tumors than in tumors with progression (*p* = 0.004). Additionally, a pre-GKRS ADCmax cutoff of 1.274 × 10^−3^ mm^2^/s predicted tumor regrowth with moderate accuracy (AUC = 0.68, sensitivity 69.2%, specificity 70%).

### 3.8. Correlation Between Clinical Parameters and Post-GKRS Growth Rates

We categorized studies based on their tumor progression definitions and summarized the prognostic findings from each study in Table 2, indicating whether a correlation was reported between each factor and tumor regrowth.

### 3.9. Complication and Sequelae

Post-treatment complications following GKRS for VS have been reported in multiple studies and are summarized in Table 3 and Figure 2. Notably, pseudo-progression was observed in 97 out of 187 patients in one study [15], a higher incidence than previously recognized.

## 4. Discussion

Vestibular schwannomas continue to present significant challenges in neurosurgery due to their unpredictable behavior and potential impact on cranial nerve functions. Previous studies have already demonstrated potential predictors in untreated VS and in VS after resection [18,19,20]. This review systematically analyzed the role of patient and tumor characteristics, clinical manifestations, and imaging biomarkers in predicting VS growth after Gamma Knife surgery. These results further emphasize the complexity of predicting tumor progression following GKRS, highlighting the various factors that are potentially involved in this regrowth.

Patient-specific characteristics such as age, gender, and administered radiation dose have demonstrated limited predictive value regarding post-GKRS tumor control. Studies have consistently shown that these factors are not significantly associated with post-GKRS tumor growth in VS [9,10,12,13,14,15,16,17]. This is consistent with analyses of their roles in the growth of untreated VS and in the growth after microsurgical resection [19,20]. Therefore, relying solely on these demographics may not suffice for prognostication or tailoring treatment strategies.

Although not investigated in this review, patient lifestyle factors including smoking, obesity, and comorbidities such as diabetes and hypertension may also affect treatment outcomes. These factors have been shown to be associated with prognosis in other studies of intracranial tumors [21,22]. It would be valuable for future VS cohorts to record and analyze such clinical cofactors.

Tumor-related factors offer a nuanced perspective. Initial tumor size might be a crucial parameter in predicting the outcomes of VS treated with GKRS. Stijn et al. [17] showed an 80.7% control rate for lesions exceeding 6 cm^3^, while Mezry et al. [23] reported 78.6% radiological control in tumors > 10 cm^3^, both notably lower than the 90% control rates achieved in smaller VS (<3 cm) as reported by current reviews [24]. These results may be related to the irregular morphology of large tumors or insufficient radiation coverage. Large tumors usually have more complex vascularity and tissue heterogeneity, which may lead to uneven distribution of radiation dose within the tumor, thereby reducing the control rate [25]. In addition, large tumors may contain more hypoxic areas, and hypoxic cells are less sensitive to radiation [26], which may be a potential mechanism for the decreased control rate. This necessitates caution when considering GKRS for a VS tumor with a large initial size.

While pre-treatment growth rates have been considered important, their predictive ability varies across studies [11,12,16]. Of note, rapidly growing VS may have other characteristics that may contribute to tumor growth, such as tumor size upon radiation, location, and cystic components, which may affect its radiosensitivity. As the current evidence is inconclusive, pretreatment growth should be interpreted in the context of other tumor features rather than as an isolated predictor.

The anatomical location of the tumor, particularly extensions into the ear canal, has also been implicated in influencing post-treatment regrowth in one study [11], possibly due to the difficulty of GKRS reaching such locations, or because extra-canalicular tumors are more invasive [20]. However, the remaining studies [9,10,12,13,14] reported no significant difference in the control rate between Gamma Knife treatment for tumors inside and outside the internal auditory canal, and no significant correlation between tumor location and tumor regrowth. In summary, the anatomical location may not be a strong predictor of tumor growth, but it may suggest a risk of more aggressive behavior.

Cystic VSs are commonly regarded as having rapid growth and unpredictable biological behavior, often leading to rapid worsening of the patient’s status [27]. However, it tends to exhibit more pronounced regression post-GKRS. Chih-Chun Wu [15] reported that compared with solid VSs, cystic VSs were more likely to regress or stabilize in the initial post-GKRS 6–12 months and at follow-up. A similar conclusion was also reached in the study by Bowden G. et al. [28], who found that a volumetric reduction of >20% occurred in 85.4% of macro-cystic tumors, 76.1% of microcystic tumors, and 62.8% of homogeneously enhancing VS. However, these findings contrast with the current systematic review associations of cystic VS control rate [29], which showed that radiosurgery for cystic VS exhibits effective tumor control probabilities similar to those for solid VS. This is similar to the three studies we included in this review [9,14,17]. This suggests that the cystic component may not be a potential predictor of long-term control outcome. While cystic lesions may produce more pronounced initial shrinkage, clinicians should not assume that cystic composition alone confers a better prognosis. Both cystic and solid VS require vigilant long-term monitoring, and any temporary early regression should be interpreted cautiously.

VS typically manifests through a spectrum of auditory and vestibular symptoms attributable to its origin from the vestibulocochlear nerve. These symptoms often serve as the initial indicators prompting diagnostic investigations such as audiometric testing and magnetic resonance imaging (MRI), leading to the detection of VS, but symptom severity does not seem to correlate with post-GKRS tumor behavior. Symptoms are more useful for staging and patient counseling than for the prediction of regrowth. Although the reviewed studies did not find symptoms to predict GKRS efficacy [10], it may be insightful to study whether symptomatic improvement, such as hearing gain or tinnitus relief, correlates with better control.

Advanced imaging biomarkers have shown promise in pre-treatment assessment. Studies have demonstrated a positive correlation between histology and imaging [30]. Elevated ADC values may reflect higher extracellular water content and a greater proportion of Antoni B tissue, which in turn indicates a higher cystic proportion [31] and could correlate with increased radiosensitivity. Previous studies have shown that MRI texture features are also positively associated with tumor regrowth after GKRS, but since some of the included patients were treated with resection or only intra-canalicular tumors were selected, they were not included in this study [32,33,34]. The potential of MRI imaging to predict tumor progression or pseudo-progression offers a valuable tool for clinicians, aiding in patient stratification and individualized treatment planning.

A thorough understanding of the possible sequelae facilitates early detection and intervention, ultimately enhancing patient outcomes and preserving neurological function. The complications reported in this paper reflect the findings in other reviews, such as that by Oscar Persson et al. [35], which showed 3.6% for facial nerve deterioration and 6.0% for trigeminal nerve deterioration following GKRS. Paolo De Sanctis et al. [36] documented that the incidence of HCP was 3% (95% CI 2–4%) after Gamma Knife radiosurgery, with surgical cerebrospinal fluid diversion required in 88% of these patients. The occurrence of complications highlights the need for post-treatment monitoring. Pseudo-progression, in particular, needs careful interpretation to distinguish it from true tumor progression, thereby preventing unwarranted interventions and anxiety in patients.

In our review, the included studies reported a wide range of marginal doses (8.3–20.0 Gy) (see Appendix A), although no statistically significant association was reported between marginal radiation dose and post-GKRS tumor progression, and the considerable variation in prescribed doses across studies represents a methodological source of heterogeneity. This variation may influence indirect radiobiological responses or follow-up interpretations, thereby complicating direct comparisons of tumor control rates. Higher doses may improve local control but increase the risk of complications, such as cranial nerve dysfunction, whereas lower doses may be less effective for larger or rapidly growing tumors [37,38]. In addition, differences in treatment planning may also affect treatment effects. This variability limits direct comparisons between studies. Prospective, multicenter studies with standardized protocols and longer follow-up are needed to validate and refine these predictors for clinical use.

Meanwhile, future directions should focus on integrating molecular and genetic profiling to unravel the underlying mechanisms driving tumor behavior. Understanding the role of specific growth factors and signaling pathways could pave the way for targeted therapies, enhancing the effectiveness of existing treatment options. Furthermore, advanced imaging techniques and biomarkers could refine diagnostic precision and prognostic accuracy. Techniques such as DTI, perfusion-weighted imaging (PWI), and positron emission tomography (PET) could provide deeper insights into the tumor microenvironment, vascularity, and metabolic activity. The utilization of these imaging biomarkers may enhance the ability to predict tumor response to GKRS and identify early signs of treatment failure or pseudo-progression.

### Limitations

The current understanding of growth determinants in VS treated with GKRS is constrained by several limitations inherent in the existing literature. Foremost, nearly all reviewed studies were retrospective case series, introducing bias from non-random patient selection and heterogeneous imaging follow-up. Treatment parameters varied across studies, not only in radiation dose, but also in radiosurgical technique and follow-up intervals. Critically, there is no consensus definition of tumor progression: some series used a linear growth criterion (>2 mm increase), while others applied volumetric thresholds (>10–20%). Such inconsistencies in outcome measures hinder data pooling and comparison. Finally, our analysis is inherently limited to reported factors; unmeasured influences could not be assessed.

## 5. Conclusions

The growth of VS after GKRS is affected by multiple factors. For patients who receive Gamma Knife radiosurgery, patient-specific factors have limited predictive effects on treatment outcomes. However, factors including initial tumor volume, pretreatment growth rate, and MRI features may provide more reliable prediction of growth post-GKRS. Prospective validation in larger, multicenter cohorts and integration with clinical, imaging biomarkers, and molecular data are needed to establish robust, widely applicable prediction models.

## Figures and Tables

**Figure 1 cancers-17-01993-f001:**
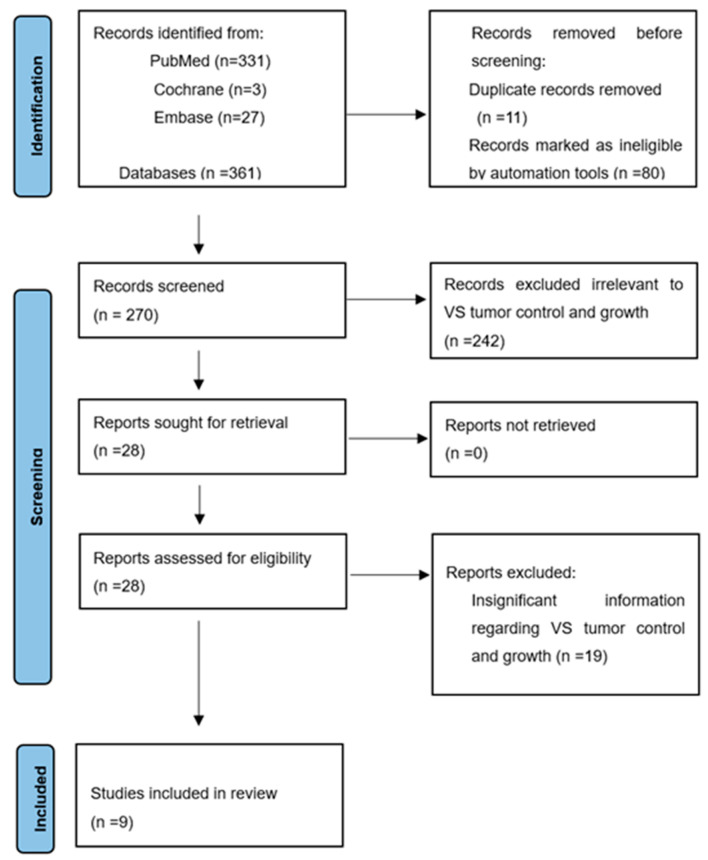
PRISMA 2020 diagram showing the inclusion process.

**Figure 2 cancers-17-01993-f002:**
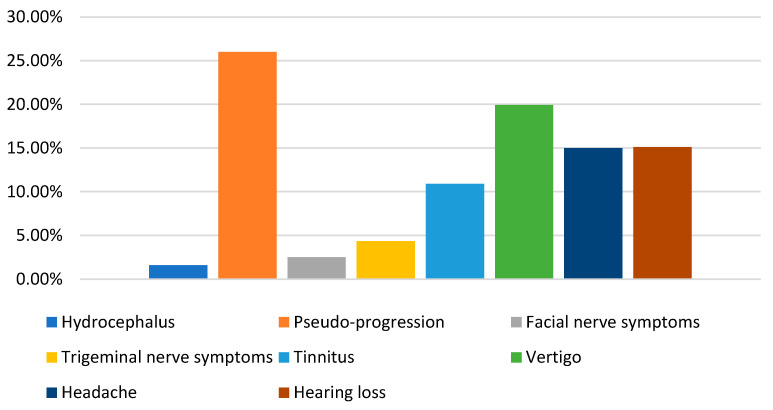
New or increased complications/sequelae.

**Table 1 cancers-17-01993-t001:** General patient characteristics included in the study.

Study Number	Author, Year	Type	Number of Patients	Age (Year)	Tumor Control	Length of Follow-Up (mo)
1	Grzegorz Turek [9], 2023	Gamma Knife	94	55	87.2%	41 (6–69)
2	Ferdinand C. A. Timmer [10], 2011	Gamma Knife	100	57 (25–85)	92%	26 (24–48)
3	Alexander P. Marston [11], 2017	Gamma Knife	68	67 (23–88)	86.8%	43.5 (14–147)
4	Theresa Wangerid [12], 2014	Gamma Knife	128	64 (23–89)	92%	86 (5–170)
5	Stephen Johnson [13], 2019	Gamma Knife	871	57 (18–95)	97%	62.4 (96–240)
6	Rick van de Langenberg [14], 2011	Gamma Knife	33	54.8 (30–83)	88%	30 (12–72)
7	Chih-Chun Wu [15], 2017	Gamma Knife	187	52.2 (20.4–82.3)	90.9%	60.8 (24–128.9)
8	Soroush Larjani [16], 2014	Gamma Knife	63	64 (26–83)	88.9%	32 (12–72)
9	Stijn Klijn [17], 2016	Gamma Knife	420	57.6 ± 12.7	89.3%	61.2 (48.0–84.0)

This table displays the raw data of nine included studies. It displays the number of patients, age, tumor control rate, and follow-up length included in each study. All the studies used Gamma Knife radiosurgery.

**Table 2 cancers-17-01993-t002:** Significant prognostic factors reported in the included studies by stratified analysis.

Author, Year	Growth Rate Definition	Gender	Age	Radiation Dose	Initial Tumor Size	Tumor Growth Rate Before GKRS	Location	Cystic Exist	Clinical Presentation	MRI
Theresa Wangerid [12], 2014	Linear > 2 mm	/	×	/	/	×	×	/	/	/
Stijn Klijn [17], 2016	Linear > 2 mm	/	×	×	√	/	/	×	/	/
F.C.A. Timmer [10], 2011	Linear > 2 mm	×	×	×	×	/	×	/	×	/
Alexander P. Marston [11], 2017	Linear > 2 mm	/	/	/	×	√	√	/	/	/
Grzegorz Turek [9], 2023	Volume > 10%	×	×	×	/	/	×	×	/	/
Stephen Johnson [13], 2019	Volume > 15%	/	×	×	√	/	×	/	/	/
Chih-Chun Wu [15], 2017	Volume > 10%	×	×	×	×	/	/	√	/	√
Soroush Larjani [16], 2014	Volume > 20%	×	×	/	×	×	/	/	/	/
Rick van de Langenberg [14], 2011	No need for further treatment	×	×	/	×	/	×	×	/	/

/: Not mentioned. √: Correlation. ×: Not correlation. Volume: An increase in tumor volume compared to baseline volume on follow-up imaging. Linear: An increase in the maximum diameter of the tumor on follow-up imaging. This table shows all the predictive factors associated with tumor regrowth after GKRS in the included studies by stratified analysis, where √ represents a study that confirmed an association, × represents a study that confirmed no association, and / represents no subgroup study.

**Table 3 cancers-17-01993-t003:** New or increased complications/sequelae.

New or Increase Complication/Sequelae	No. of Patients
Hydrocephalus (HCP)	12/768 (1.6%)
Pseudo-progression	133/510 (26%)
Facial nerve symptoms	43/1709 (2.52%)
Trigeminal nerve symptoms	67/1546 (4.33%)
Tinnitus	169/1548 (10.9%)
Vertigo	135/677 (19.94%)
Headache	15/100 (15%)
Hearing loss	5/33 (15.1%)

This table displays the newly occurred or increased complications or sequelae from the nine included studies. The first column lists the complications, and the second is the probability of occurrence, where the denominator is the total number of patients in the studies that mentioned this complication and the numerator is the number of people who developed the complication.

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
