# Peer review of "Predictors of Growth of Vestibular Schwannoma After Gamma Knife Treatment: A Systematic Review"

_cancers, 2025, doi:10.3390/cancers17121993_

Round 1
Reviewer 1 Report
Comments and Suggestions for Authors
This manuscript presents a systematic review aiming to identify predictive factors for vestibular schwannoma (VS) recurrence after Gamma Knife treatment. While the topic is clinically relevant, there are several methodological concerns that should be addressed.
All included studies employed Gamma Knife radiosurgery, but the radiation protocols—such as marginal dose and treatment planning—varied considerably across studies. The authors should discuss how these differences in treatment delivery might affect tumor control rates and the comparability of outcomes. Moreover, the title currently uses the term “radiation therapy,” which may be too broad; “Gamma Knife treatment” would be more accurate.
Although the review appropriately notes that definitions of tumor progression differed among studies (e.g., linear vs. volumetric criteria), it does not clearly explain how this variability was handled in evaluations. Without addressing this heterogeneity, the validity of any overarching conclusions is weakened. A stratified analysis or more explicit acknowledgment of this limitation would improve the quality of the review.
The authors mention that missing data were marked as “not applicable,” but there is insufficient explanation of how studies lacking MRI-based metrics (such as ADC values) were incorporated into the analysis.
Finally, the statistical approach is not well-described. No meta-analysis was conducted, and there is no mention of heterogeneity testing, sensitivity analyses, or evaluation of publication bias. The absence of these components raises concerns about the strength of the conclusions and should be addressed more thoroughly.
Author Response
Dear Reviewer,
We would like to sincerely thank you for your constructive and detailed feedback on the methodology of our manuscript. Please kindly find our point-by-point responses below:
- Comment: Authors should discuss how radiation protocol differences in treatment delivery might affect tumor control rates and the comparability of outcomes.
Response: We appreciate this important suggestion. In response, we have added a paragraph to the Discussion section (Line 330-341) that addresses the potential impact of differences in radiation dose and treatment planning on tumor control rates and inter-study comparability. We also created a new Supplementary Table (Table S2) to summarize key radiotherapy parameters across the included studies. This allows readers to better assess the variability in treatment protocols and their implications for data interpretation.
- Comment: The title currently uses the term “radiation therapy,” which may be too broad; “Gamma Knife treatment” would be more accurate.
Response: Thank you for this suggestion. We agree that “Gamma Knife treatment” is a more accurate and specific term. Accordingly, we have revised the manuscript title to: “Predictors of Vestibular Schwannoma Growth after Gamma Knife Treatment: A Systematic Review.”
In addition, we have replaced all “radiotherapy” in the manuscript with “Gamma Knife radiosurgery” or “GKRS” depending on the context, to maintain clarity and consistency.
- Comment: It does not clearly explain how tumor progression definitions variability was handled in evaluations.
Response: We acknowledge that this was not sufficiently addressed in the original submission. To improve clarity, we have introduced a new subsection in the Results section titled “3.8 Correlation Between Clinical Parameters and Post-GKRS Growth Rates” and we modified Table 2 to categorize studies based on their tumor progression definitions and summarized the prognostic findings from each study, indicating whether a correlation was reported between each factor and tumor regrowth. Additionally, we expanded the Limitations section to acknowledge that heterogeneity in the definition of progression affects the comparability of results and the interpretation of predictive factors.
- Comment: There is insufficient explanation of how studies lacking MRI-based metrics (such as ADC values) were incorporated into the analysis.
Response: We thank the reviewer for this insightful observation and apologize for the confusion caused by our wording in the original manuscript. The phrase “missing data were reported as not applicable” was poorly chosen and may have given the impression that we attempted to include studies that were not applicable.
To clarify, among the nine included studies, only one reported MRI-based metrics, the remaining studies did not mention it. However, these studies were retained for the synthesis of clinical and anatomical prognostic factors.
To address this issue and prevent this misunderstanding, we have revised the sentence in the Methods section (2.3) to more accurately reflect our approach: “If data or specific prognosis factors were not reported in an included study, we marked the corresponding cells in the data extraction table with a forward slash (‘/’) to indicate non-mentioned. ”
We hope this correction resolves the ambiguity and better conveys our data handling methodology.
- Comment: The statistical approach is not well-described.
Response: We appreciate your comments. We revised the Methods section (section 2.4) to explain that during the initial data extraction and study assessment phase, we did not conduct a meta-analysis but rather a qualitative analysis due to significant methodological heterogeneity in outcome definitions and radiotherapy regimens among the included studies, and explained our approach to assessing heterogeneity, sensitivity, and publication bias. In the Results section (section 3.2), although a formal statistical assessment was not possible, we conducted a qualitative assessment of heterogeneity, sensitivity, and publication bias.
We would like to once again express our sincere gratitude to the reviewer for the thoughtful and constructive feedback. Your comments have significantly improved the clarity, accuracy, and overall quality of our manuscript. We truly appreciate your time and effort in reviewing our work.
Best wishes,
Cheng Yang, MD
Yasin Temel, MD, PhD
Maastricht University Medical Center+
Reviewer 2 Report
Comments and Suggestions for Authors
The authors present an interesting review about predictive factors of growth after radiosurgery for vestibular schwannoma. 9 studies were identified , which fullfilled their search criteria. Although results were not consistent overall, the literature review provides a good overview over published data so far.
Some minor typos need to be corrected.
Author Response
Dear Reviewer,
We sincerely thank the reviewer for the positive and encouraging comments regarding the overall quality and relevance of our systematic review. In response to your suggestion, we have carefully proofread the entire manuscript and corrected all identified typographical and grammatical errors. We hope that the revised version meets your expectations.
Thank you again for your helpful feedback, and hope that this article can further clarify the current understanding and management of VS.
Best wishes,
Cheng Yang, MD
Yasin Temel, MD, PhD
Maastricht University Medical Center+
Reviewer 3 Report
Comments and Suggestions for Authors
This paper reports on factors affecting and predicting acoustic neuroma regrowth after stereotactic radiosurgery. The work is a literature review and the methods of research is good and well described and represented by PRISMA diagram. It's surprising that only 9 papers have been considered acceptable for the review. A total of 1964 cases are included. Results are well described as well as statistical analysis is correct. Introduction and discussion are satisfactory. Overall, the manuscript don't contains relevant results as reported associations with tumor regrowth are in general well know. The paper underline the reduced actual knowledge about this factors in terms of statistical and evidence based data. So, the paper is in the interest ion general audience. I have not particular criticism but I consider that Authors have to focus this particular aspect suggesting further investigation. As SRS for both newly diagnosed and post-surgery residual acoustic neuroma is becoming more and more utilized, a prospective multi centric study should be advocated in discussion and conclusion.
Author Response
Dear Reviewer,
We thank the reviewer for the constructive feedback and for recognizing the methodological quality and clinical relevance of our review. Since only studies with SRS as the first-choice treatment option were included, only 9 papers were considered acceptable for review. Meanwhile, we appreciate your observation that, although many of the reported associations with tumor regrowth are already known, the study highlights the lack of robust, statistically powered, and evidence-based data in this area. This aligns with one of our key messages that the current literature lacks standardization and prospective validation.
In response to your valuable suggestion, we have revised both the Discussion and Conclusion sections to explicitly emphasize the urgent need for prospective, multicenter studies and advocate for future investigations that incorporate consistent tumor progression criteria and establish robust, widely applicable prediction models to strengthen the predictive framework for tumor control after SRS.
We are grateful for your feedback and believe that your input has improved the clarity and impact of our manuscript.
Best wishes,
Cheng Yang, MD
Yasin Temel, MD, PhD
Maastricht University Medical Center+
Round 2
Reviewer 1 Report
Comments and Suggestions for Authors
The authors responded sincerely to the reviewers' questions and made considerable revisions. Their approach of clearly stating the limitations rather than forcing analysis with incomplete data and conducting analysis and interpretation to the best of their ability is commendable. The current version is worthy of acceptance.